

# Exogenous melatonin improves salt tolerance mainly by regulating the antioxidant system in cyanobacterium *Nostoc flagelliforme*

Xiaolong Yuan[1], Jing An[1], Tao Zheng[1] and Wenjian Liu[2]

[1] School of Food and Biological Engineering, Shaanxi University of Science & Technology, Xi'an, China
[2] School of Environmental Science and Engineering, Shaanxi University of Science & Technology, Xi'an, China

## ABSTRACT

Melatonin is a multifunctional nontoxic bio-stimulant or signaling molecule, generally distributing in different animal and plant organs for invigorating numerous physiological processes against abiotic stresses. In this study, we investigated the potential impact of melatonin on the cyanobacterium *Nostoc flagelliforme* when exposed to salt stress according to some biochemical and physiological parameters, such as relative electrolyte leakage, PSII activity, and photosynthetic pigments including chlorophyll a, phycocyanobilin, and phycoerythrobilin. We found that melatonin could also maintain $K^+$ homeostasis in salt-stressed *N. flagelliforme*. These above results confirmed melatonin had multiple functions in hyperosmotic stress and ion stress caused by salinity. Notably, we observed melatonin could regulate the reactive oxygen species (ROS) signal and distinctly decrease the content of hydrogen peroxide and superoxide anion in salt-stressed cells, which were largely attributed to the increased antioxidant enzymes activities including catalase, superoxide dismutase, ascorbate peroxidase, and glutathione reductase. Finally, qRT-PCR analysis showed that melatonin stimulated the expression of antioxidant genes (*NfCAT*, *NfSOD*, and *NfGR*). In general, our findings demonstrate melatonin has beneficial effects on *N. flagelliforme* under salt stress by intensively regulating antioxidant system.

## INTRODUCTION

Melatonin (N-acetyl-5-methoxytryptamine) is a ubiquitously distributed molecule that exists in both animals, plants, and microorganisms (*Hardeland et al., 2010*; *Kanwar, Yu & Zhou, 2018*; *Xie et al., 2022*), acts as a pleiotropic signaling molecule and plays multifunctional roles in seed germination, root, and shoot development (*Tan et al., 2012*; *Park & Back, 2012*; *Martinez et al., 2018*). Melatonin is usually involved in signal transduction under salt stress including hydrogen peroxide and abscisic acid (*Zhang et al., 2014*; *Gong et al., 2017*). From last few years, melatonin has been identified as a novel class of metabolic regulator in the biological kingdoms. It can improve the tolerance of plants to

Corresponding author
Jing An, anjing@sust.edu.cn

numerous abiotic stresses such as high temperature (*Shi et al., 2015b*), cold (*Li et al., 2017b*), salt (*Jiang et al., 2016*; *Siddiqui et al., 2019*; *Zhan et al., 2019*), and heavy metal (*Posmyk et al., 2008*). The endogenus melatonin levels show the circadian rhythm, indicating a role in photoperiodic and rhythmic phenomena in the dinoflagellate unicellular algae *Gonyaulax polyedra* (*Caniatio et al., 2003*). In addition, melatonin can effectively scavenge ROS such as $H_2O_2$ and $O_2^{\bullet-}$ in crops, vegetables, and algae (*Wei et al., 2015*; *Shi et al., 2015a*; *Ke et al., 2018*; *Zhang et al., 2018*). It has been demonstrated that exogenous melatonin promotes the tolerance against adverse environmental factors through detoxifying ROS up to tolerable levels by enhancing antioxidant enzymes activities and regulating the transcription levels of genes related to the antioxidant system in organisms (*Rodriguez et al., 2004*; *Fischer et al., 2013*). The main mechanism by which melatonin enhances the tolerance of diverse abiotic stresses is attributed to its antioxidative role as a scavenger of reactive oxygen species (ROS) (*Weeda et al., 2014*).

Many areas of the world, particularly arid and semi-arid regions, are severely affected by soil salinization (*Himabindu et al., 2016*). The accepted view is that high salinity causes multiple stresses (hyperosmotic, ionic, and oxidative) which limit the growth and development of plants (*Carillo et al., 2019*). At present, a large number of studies have focused on osmotic stress and ion stress caused by salinity. The excessive NaCl in the soil can directly cause inconvenience to water use efficiency due to hyperosmotic stress (*Munns & Tester, 2008*). In addition, the NaCl-induced ionic stress disturbs the $K^+$/$Na^+$ balance (*Gao et al., 2016*), reduces the accumulation of photosynthetic pigments (*Kao, Tsai & Shih, 2003*), and disrupts the enzymatic reaction processes (*Li et al., 2012*). It is noteworthy that the oxidative stress caused by NaCl should not be ignored, oxidative stress is the most dangerous event under salt conditions which triggers overproduction of ROS that cause extensive cellular death and DNA damage, collapse the enzymatic actions, along with distraction of the antioxidant defense system and dysfunction of physiological and molecular mechanisms of plants (*Mittler, 2002*; *Apel & Hirt, 2004*; *Miller et al., 2010*; *Caverzan, Casassola & Brammer, 2016*; *Choudhary, Kumar & Nirmaljit, 2020*). Thus, more attention should be payed on the damage of oxidative stress to plants or other organisms under high salinity.

Most studies have confirmed the beneficial effects of melatonin on plants after exposure to salt stress. However, its potential effects on cyanobacterium have been less understood. *Nostoc flagelliforme* is a terrestrial and nitrogen-fixing cyanobacteria that grows in arid and semi-arid steppes (*Gao, 1998*). It can change the nutrient composition and water conduction ability of the sand by fixing the soil particles (*Chen et al., 2011*). These functions can affect the distribution of vegetation and turn desert into fertile soil (*Bailey, Mazurak & Rosowski, 1973*). Unfortunately, *N. flagelliforme* often suffer from the harsh environment, especially salt stress (*Ye & Gao, 2004*). Therefore, alleviating the damage of salt stress to *N. flagelliforme* can improve its application value and protect the ecological environment. In microalgae, some reports have proved exogenous melatonin could alleviate abiotic stresses including salt (*Zhao et al., 2021*) and high light (*Ding et al., 2018*). However, the application of melatonin on alleviating salt stress to the macroscopic cyanobacterium *N. flagelliforme* has never been reported.

In this study, we clarified the powerful role of melatonin in *N. flagelliforme* to adapt to salt stress from the physiological and molecular levels. We found exogenous melatonin could facilitate the photosynthetic pigments synthesis and maintain $K^+$ homeostasis under salt stress. Moreover, application of melatonin significantly reduced the content of $H_2O_2$ and $O_2^{\bullet-}$ in *N. flagelliforme* under salt stress and the corresponding antioxidant enzymes activities were increased. qRT-PCR analysis verified the molecular fact that melatonin could enhance the salt tolerance of cyanobacteria *N. flagelliforme*. Our findings emphasized the antioxidant roles of melatonin against salt stress in *N. flagelliforme*. We firstly explained the regulatory mechanism of melatonin to resist salt stress, which could be useful for improving the eco-friendly terrestrial cyanobacterium to adapt to high-salinity habitats.

# MATERIALS AND METHODS

## Cyanobacterium, culture conditions, and experimental treatments

The terrestrial cyanobacterium *Nostoc. flagelliforme* (Yinchuan, Northwest of China) was used in this study. Algal cells were cultured in 250 mL glass flasks containing 100 mL Blue Green-11 (BG-11) medium at 25 °C under continuous cool-white fluorescent light illumination of 40 μmol photons $m^{-2}$ $s^{-1}$ (*Ai et al., 2014*). The flasks were shaken at 130 rpm in a shaker. The 100 mL BG-11 medium was composed of $K_2HPO_4$ (0.23 mM), $MgSO_4 \cdot 7H_2O$ (0.30 mM), $CaCl_2 \cdot 2H_2O$ (0.24 mM), $C_6H_8O_7$ (0.03 mM), $C_6H_{10}FeNO_8$ (0.02 mM), EDTA disodium (3.42 μM), $Na_2CO_3$ (1.99 μM), $H_3BO_3$ (46.26 μM), $MnCl_2 \cdot 4H_2O$ (9.15 μM), $ZnSO_4 \cdot 7H_2O$ (1.38 μM), $Na_2MoO_4 \cdot 2H_2O$ (1.63 μM), $CuSO_4 \cdot 5H_2O$ (0.32 μM), $Co(NO_3)_2 \cdot 6H_2O$ (0.17 μM), and $NaNO_3$ (16.94 mM). The present experiment was divided into four groups: (i) control (BG-11 medium), (ii) salt stress (+300 mM NaCl), (iii) melatonin (+200 μM MT), (iv) melatonin and salt stress (+200 μM MT+300 mM NaCl). The algal cells were cultivated for 14 days.

## Evaluation of algal cell growth

The dry weight (DW) was used to measure the growth of *N. flagelliforme*. A total of 5 mL culture was centrifuged (8,000 rpm, 10 min) and the pellet was dried at −80 °C until the weight was constant. The relative electrolytic leakage (REL) was determined using the conductivity meter (Shanghai Instruments Inc, Shanghai, China). A total of 5 mL cell suspension was rinsed with distilled water, and the conductivity of distilled water was measured as $C_w$. The suspension was allowed to stand for 10 min and its conductivity was detected as $C_1$. Then the sample was boiled for 30 min to measure conductivity at 25 °C as $C_2$. The REL was calculated using the following equation:
REL(%) = $(C_1-C_w)/(C_2-C_w) \times 100\%$ (*An et al., 2017*).

## The photosynthetic pigments measurements

The chlorophyll a (Chl a) content was measured using a UV-vis spectrophotometer (Shanghai Spectrum Instruments Inc., Shanghai, China). A total of 5 mL cell suspension was centrifuged (8,000 rpm, 10 min) and then Chl a was extracted with 5 mL 95% ethanol for 24 h at 4 °C in the dark. After centrifuging (6,000 rpm, 5 min), the supernatant was

detected in absorbance at 664.1 and 648.6 nm, respectively. The concentration of Chl a was calculated as follows: Chl a (μg/mL) = $13.36 \times A_{664.1} - 5.19 \times A_{648.6}$ (*Zhao et al., 2008*).

The phycobiliprotein content was measured using a UV-vis spectrophotometer (Shanghai Spectrum Instruments Inc., Shanghai, China). A total of 10 mL cell suspension was centrifuged (6,000 rpm, 5 min). The pellet was washed twice with distilled water and centrifuged (6,000 rpm, 5 min). The cells were dissolved in 3 mL phosphate buffer (0.1 mM, pH 7.0) to sonicate. Then the sample was repeatedly frozen and thawed three times in liquid nitrogen and then centrifuged (10,000 rpm, 5 min). The supernatant was used to detect the phycobiliprotein content. Phycocyanin (PC) and Phycoerythrin (PE) content were calculated as follows: PC (mg/mL) = $(A_{620} - 0.0474 \times A_{652})/5.34$; PE (mg/mL) = $(A_{562} - 2.41 \times PC - 0.849 \times APC)/9.62$ (*Mishra et al., 2012*).

The PS II activity (Fv/Fm) was determined at 25 °C using a portable plant efficiency analyzer (AquaPen, Czech Republic). The cell suspension was dark-adapted for 15 min before the measurement. The specific parameters were set as follows: Time = 005 s, auto data, and 100% light (*Gao et al., 2014*).

## Na+ and K+ content measurements

A total of 5 mL cell suspension was dried to constant weight at 80 °C and digested in 10 mL 68% sulfuric acid. Then the sample was adjusted up to 10 mL with 1% nitric acid. The content of $Na^+$ and $K^+$ was measured by an atomic absorption spectrometer (ZEEnit 700P; Analytik Jena, Jena, Germany) (*Gao, Ren & Lu, 2006*).

## Determination of hydrogen peroxide and superoxide anion levels

The $H_2O_2$ content was measured by the titanium sulfate colorimetric method (*Deadman, Hellgardt & Hii, 2017*). Hydrogen peroxide reacted with titanium sulfate to produce yellow peroxide-titanium complexes, which were soluble in acid. The absorbance of the mixture can be detected at 412 nm. The content of $H_2O_2$ was calculated from a standard curve. The generation rate of superoxide anion ($O_2^{\bullet-}$) was detected by the method of hydroxylamine oxidation in the absorbance at 530 nm (*Qiu et al., 2014*). The standard curves and detection methods were based on the assay kit (Sangon Biotech, Shanghai, China), respectively.

## Antioxidant enzymes activities measurements

The superoxide dismutase (SOD) activity was assayed using the photochemical nitroblue tetrazolium (NBT) method (*Li et al., 2011*). The catalase (CAT) activity was assayed spectrophotometrically at 240 nm. Catalase can decompose $H_2O_2$, the absorbance of mixture at 240 nm decreased with reaction time, and the rate of absorbance change was used to calculate CAT activity (*Qiu et al., 2014*). Ascorbate peroxidase catalyzed the oxidation of ascorbic acid (ASA) by $H_2O_2$. The ascorbate peroxidase (APX) activity can be calculated in absorbance at 290 nm by measuring the oxidation rate of ASA (*Jiang & Zhang, 2002*). Glutathione reductase can catalyze the dehydrogenation of NADPH to generate $NADP^+$, and glutathione reductase (GR) activity was obtained by measuring the rate of decrease in absorbance at 340 nm to calculate the rate of NADPH dehydrogenation

**Table 1 Sequences of primers used in quantitative real-time PCR.**

| Gene | Primer sequnces (5′→3′) |
| --- | --- |
| NfCAT | F: GGCAAGTGACACAGGAACC |
| | R: CCACGCTCATCTTGACCATT |
| NfSOD | F: TCGGTAGTGGCTGGTCTT |
| | R: AGTCAATGTAGTAGGCGTGTTC |
| NfGR | F: CCACCTCAACACCAGATAT |
| | R: GTTACCGCTTCCAAGTCT |
| NfEST1 | F: CTGATGGAATTGGCGTTGGA |
| | R: TTAGCGTTGTTGCGTCTGTAAT |
| NfTrKA | F: TGTGGCTTGAGTGGTATTG |
| | R: GGCGGCAGAGTCTATATTG |
| 16S-rRNA | F: CAGGTGGCAATGTAAGTCT |
| | R: TCGTCCCTCAGTGTCAGTT |

**Note:**
The gene specific primers were listed.

(*Cui et al., 2017*). All of these antioxidant enzymes activities were quantified using the enzyme assay kit (Sangon Biotech, Shanghai, China), respectively.

## RNA extraction and qRT-PCR analysis

Algal cells from different treatment groups at 0, 3, 6, 12 and 24 h were collected and stored at −80 °C. Three independent biological replicates, each containing a pool of 10 mL culture, were sampled for transcriptome analysis. Total RNA was extracted from frozen algal samples using TRIzol reagent (Tiangen, Beijing, China). Then, 1 μg total RNA of each sample was reverse transcribed according to manufacturer instructions (TUREscript 1st Strand cDNA Synthesis Kit; PIONEER, Dongguan, China). The gene specific primers were listed in Table 1; qRT-PCR was performed in 20 μL volume according to manufacturer instructions (2 × Sybr Green qPCR Mix; PIONEER, Dongguan, China). PCR cycles were set up as follows: pre-denaturation at 95 °C for 3 min, 40 cycles of 95 °C for 15 s and 60 °C for 30 s. The gene relative expression levels were normalized and calibrated according to the $2^{-\Delta\Delta CT}$ method (*Schmittgen & Livak, 2008*), with the16S-rRNA gene as the internal reference.

## Statistical analysis

All data were analyzed *via* SPSS Ver 25.0 statistical software. Values were presented as the mean ± SD ($n = 3$). Statistical differences in the data were analyzed using one-way ANOVA followed by Duncan's multiple range tests. The significance was established at the $p < 0.05$ level.

# RESULTS

## Effects of melatonin on cell growth under salt stress

The growth and appearance of algal cells after different treatments were investigated. The dry weight (DW) was significantly declined under salinity stress compared with other
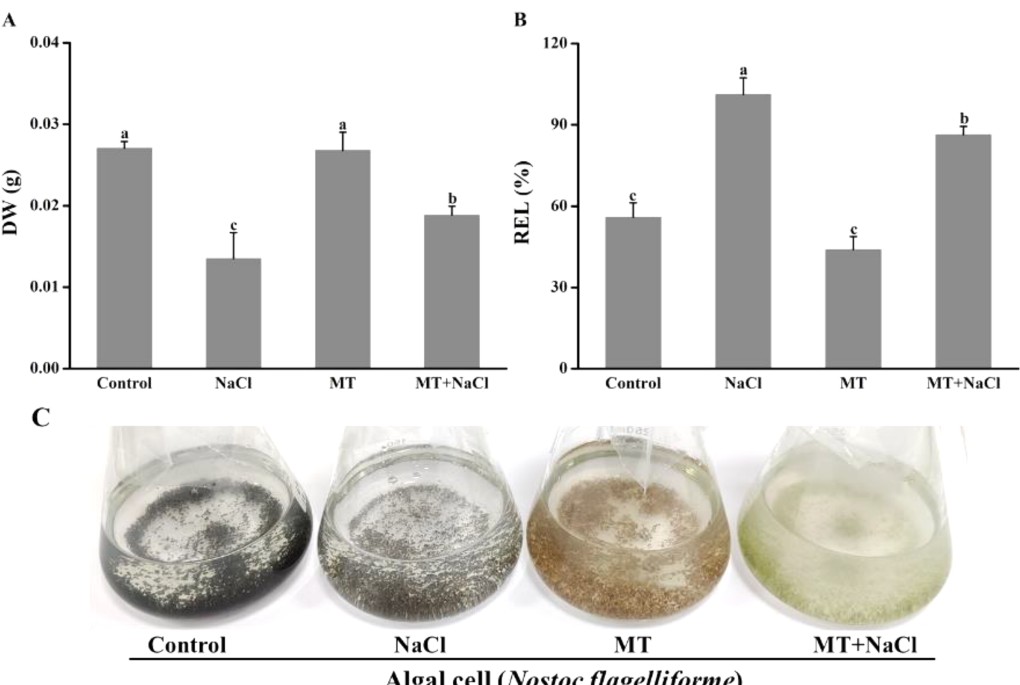

**Figure 1** **The growth and appearance of algal cells after different treatments were investigated.** Effects of melatonin on (A) dry weight (DW), (B) relative electrolytic leakage (REL), and (C) appearance of algal cells at 14 days under 300 mM NaCl stress. Data represent the mean ± SD ($n$ = 3). Columns labelled with different letters among treatments show statistical differences ($p < 0.05$).

treatments. The DW of MT-treated culture under salt stress was significantly increased compared with the only salinity. The melatonin alone did not affect DW compared with the control (Fig. 1A). The relative electrolyte leakage (REL) of salt-stressed culture was higher than other cultures. After treatment with melatonin, the REL values significantly decreased under salt stress, but the melatonin alone did not affect REL compared with the control (Fig. 1B). It indicated that melatonin could protect the permeability f cell membranes under salt stress. Meanwhile, we found algal cells showed different appearance especially application of melatonin (Fig. 1C).

## Effects of melatonin on photosynthetic pigments under salt stress

Melatonin pretreatment could change the color of algal cells, and then the impact of melatonin on photosynthetic pigments was explored. Compared with the control, the Chl a, PC, and PE content were reduced in other cultures (Figs. 2A–2C). It indicated that melatonin could decrease the Chl a, PC, and PE content with or without salt stress. Interestingly, after treatment with melatonin, Fv/Fm, which reflects the maximum quantum efficiency of photosystem II, was significantly increased under NaCl stress (Fig. 2D).

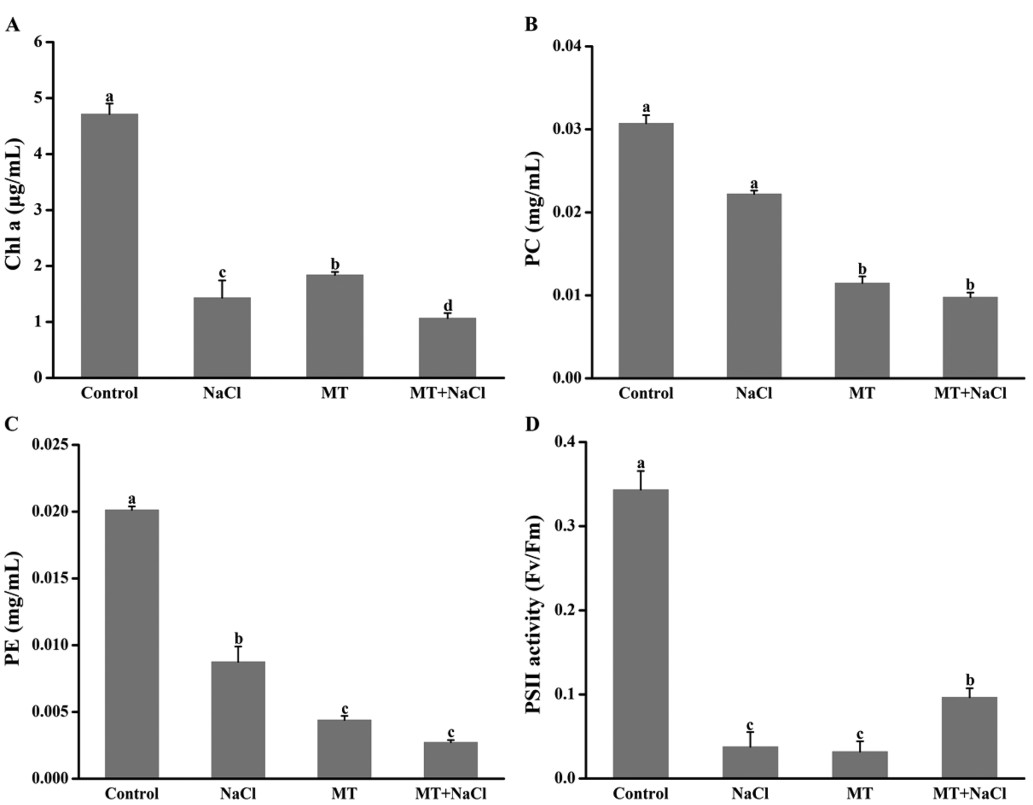

**Figure 2 Melatonin pretreatment could change the color of algal cells, and then the impact of melatonin on photosynthetic pigments was explored.** Effects of melatonin on (A) chlorophyll a (Chl a) content, (B) phycocyanin (PC) content, (C) phycoerythrin (PE) content, and (D) PSII activity of algal cells at 14 days under 300 mM NaCl stress. Data represent the mean ± SD ($n$ = 3). Columns labelled with different letters among treatments show statistical differences ($p < 0.05$).

## Effects of melatonin on Na$^+$ and K$^+$ contents under salt stress

To evaluate the effects of melatonin on K$^+$ retention under saline conditions, NaCl-induced ion contents in algal cells were detected. Melatonin treatment significantly alleviated the increased of Na$^+$ accumulation and K$^+$ exosmosis in algal cells under salt stress (Figs. 3A and 3B). The melatonin alone improved the content of Na$^+$ and declined the content of K$^+$ compared with the control, respectively. Interestingly, MT treatment contributed to the increase of the K$^+$/Na$^+$ ratio in algal cells under salt stress, but the melatonin alone did not affect the K$^+$/Na$^+$ ratio compared with the control (Fig. 3C).

## Effects of melatonin on the expression patterns of osmotic responsive and K$^+$ channel gene under salt stress

To explore the effects of melatonin on the expression of genes related to osmotic responsive and K$^+$ channels under salt stress, we determined the expression levels of *NfEST1* and *NfTrKA*. *NfEST1* is relate to osmotic stress. The relative high level expression of *NfTrKA* indicates relatively large damage to cell. *NfTrKA* is potassium channel protein gene, which cause the K$^+$ efflux resulting in the imbalance of intracellular charge.

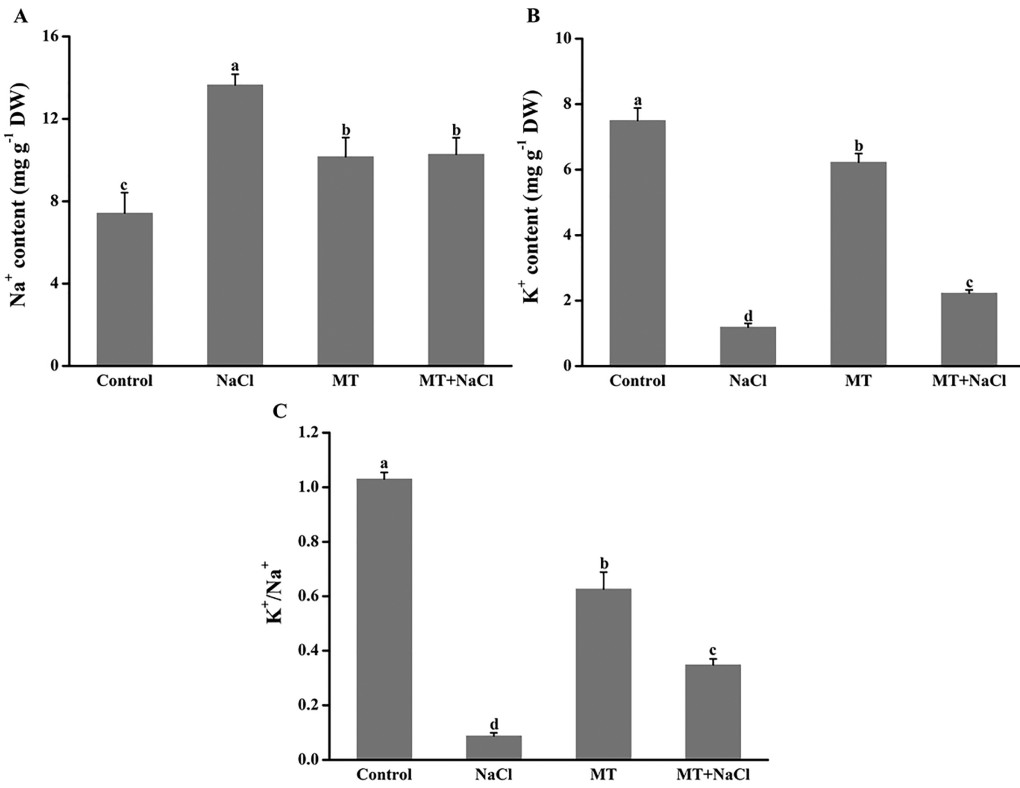

**Figure 3 To evaluate the effects of melatonin on K$^+$ retention under saline conditions, NaCl-induced ion contents in algal cells were detected.** Effects of melatonin on (A) Na$^+$ content, (B) K$^+$ content, and (C) K$^+$/Na$^+$ ratio of algal cells at 14 days under 300 mM NaCl stress. Data represent the mean ± SD ($n = 3$). Columns labelled with different letters among treatments show statistical differences ($p < 0.05$).

The expression levels of these two genes exhibited no distinction under control and melatonin culture. Salt stress significantly increased the expression levels of these genes, particularly after 6 and 12 h of salinity stress. After addition of melatonin, the expression levels of *NfEST1* and *NfTrKA* were markedly decreased under salt stress (Fig. 4).

## Effects of melatonin on H$_2$O$_2$ and O$_2^{\bullet-}$ concentration under salt stress

To investigate the role of melatonin in inhibiting ROS overproduction under salt stress, we determined the formation of H$_2$O$_2$ and O$_2^{\bullet-}$ in algal cells (Fig. 5). Excessive accumulation of hydrogen peroxide and superoxide anion under salt stress could cause oxidative stress. Compared with other treatments, the concentration of H$_2$O$_2$ and O$_2^{\bullet-}$ were remarkably increased under salt stress. However, melatonin treatment could significantly reduce the ROS levels under salt stress. The melatonin alone did not affect ROS accumulation compared to the control.

## Effects of melatonin on antioxidant enzymes activities under salt stress

To further assess the mechanism of melatonin in improving salt tolerance of algal cells, we examined the changes in antioxidant enzymes activities (Fig. 6). Compared with the

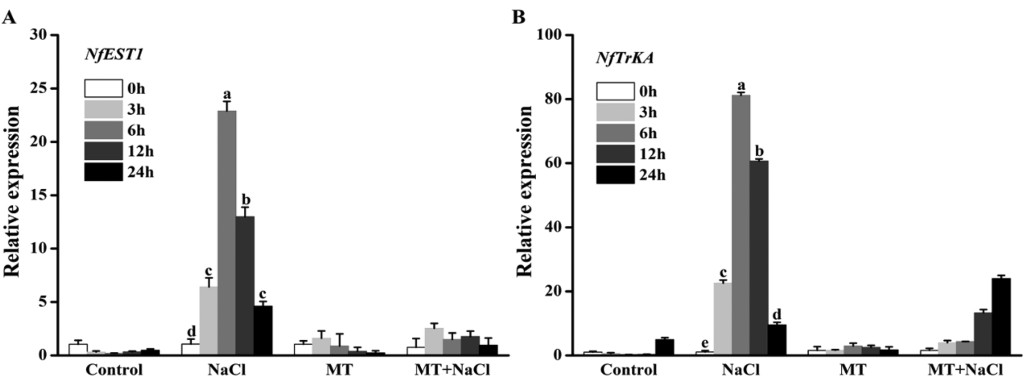

**Figure 4 To explore the effects of melatonin on the expression of genes related to osmotic responsive and K+ channels under salt stress, we determined the expression levels of *NfEST1* and *NfTrKA*.** Effects of melatonin on the expression levels of (A) *NfEST1* and (B) *NfTrKA* of algal cells under 300 mM NaCl stress. Data represent the mean ± SD ($n$ = 3). Columns labelled with different letters among treatments show statistical differences ($p < 0.05$).

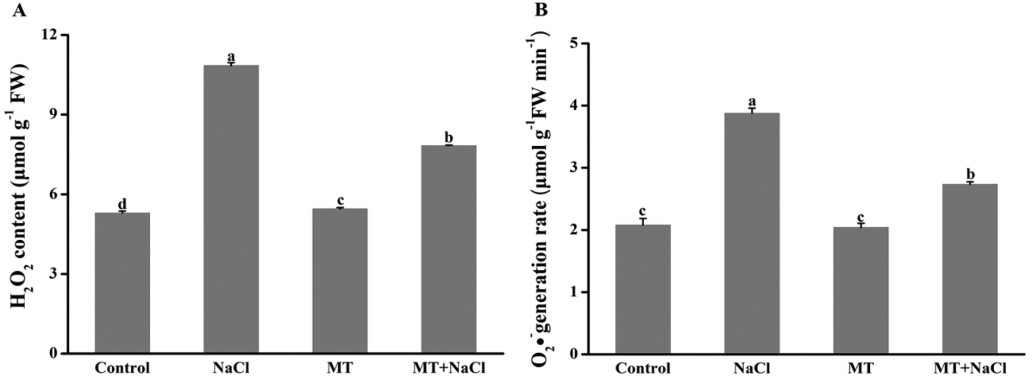

**Figure 5 To investigate the role of melatonin in inhibiting ROS overproduction under salt stress, we determined the formation of $H_2O_2$ and $O_2^{\bullet-}$ in algal cells.** Effects of melatonin on (A) $H_2O_2$ content, and (B) $O_2^{\bullet-}$ generation rate of algal cells at 14 days under 300 mM NaCl stress. Data represent the mean ± SD ($n$ = 3). Columns labelled with different letters among treatments show statistical differences ($p < 0.05$).

control, NaCl treatment exhibited the increased activities of catalase (CAT), superoxide dismutase (SOD), ascorbate peroxidase (APX), and glutathione reductase (GR) in algal cells, respectively. Application of melatonin to algal cells significantly increased the activities of these enzymes under NaCl treatment.

## Effects of melatonin on the expression patterns of antioxidant enzymes genes under salt stress

To further explore the effects of melatonin on the antioxidant defense system of *N. flagelliforme* under salt stress, we investigated the relative expression levels of antioxidant enzymes genes through qRT-PCR. NaCl treatment could increase expression

 

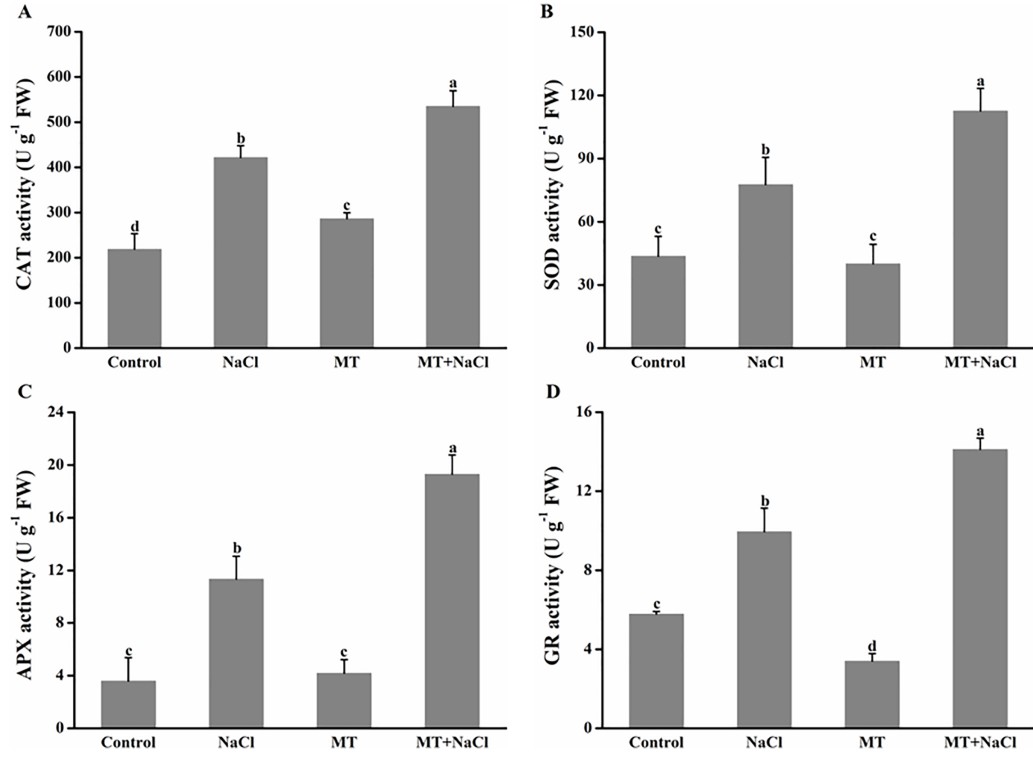

**Figure 6 To further assess the mechanism of melatonin in improving salt tolerance of algal cells, we examined the changes in antioxidant enzymes activities.** Effects of melatonin on (A) catalase (CAT), (B) superoxide dismutase (SOD), (C) ascorbate peroxidase (APX), and (D) glutathione reductase (GR) activities of algal cells at 14 days under 300 mM NaCl stress. Data represent the mean ± SD ($n = 3$). Columns labelled with different letters among treatments show statistical differences ($p < 0.05$).

levels of *NfCAT*, *NfSOD*, and *NfGR* in algal cells, respectively. After addition of melatonin, the expression of *NfCAT* was markedly increased, particularly after 3 and 6 h under salt stress (Fig. 7A); the expression levels of *NfSOD* and *NfGR* were markedly increased, particularly after 12 and 24 h under salt stress (Figs. 7B and 7C). It indicated that melatonin could increase the expression of antioxidant genes in algal cells under salt stress.

## DISCUSSION

Extensive evidence has revealed the beneficial role of melatonin in improving salt tolerance of different plant species, such as crop (*Liang et al., 2015*; *Li et al., 2017c*; *Chen et al., 2018*), fruit (*Li et al., 2017a*), and vegetable (*Wang et al., 2016*). Despite some researches proved exogenous melatonin could alleviate abiotic stresses in microalgae (*Ding et al., 2018*; *Zhao et al., 2021*), there is no recent study to understand how melatonin alleviates the damage of salt stress to eco-friendly terrestrial cyanobacterium. In the current study, we for the first time report exogenous melatonin can improve salt tolerance of terrestrial cyanobacterium *N. flagelliforme*, particularly regulating the antioxidant system.
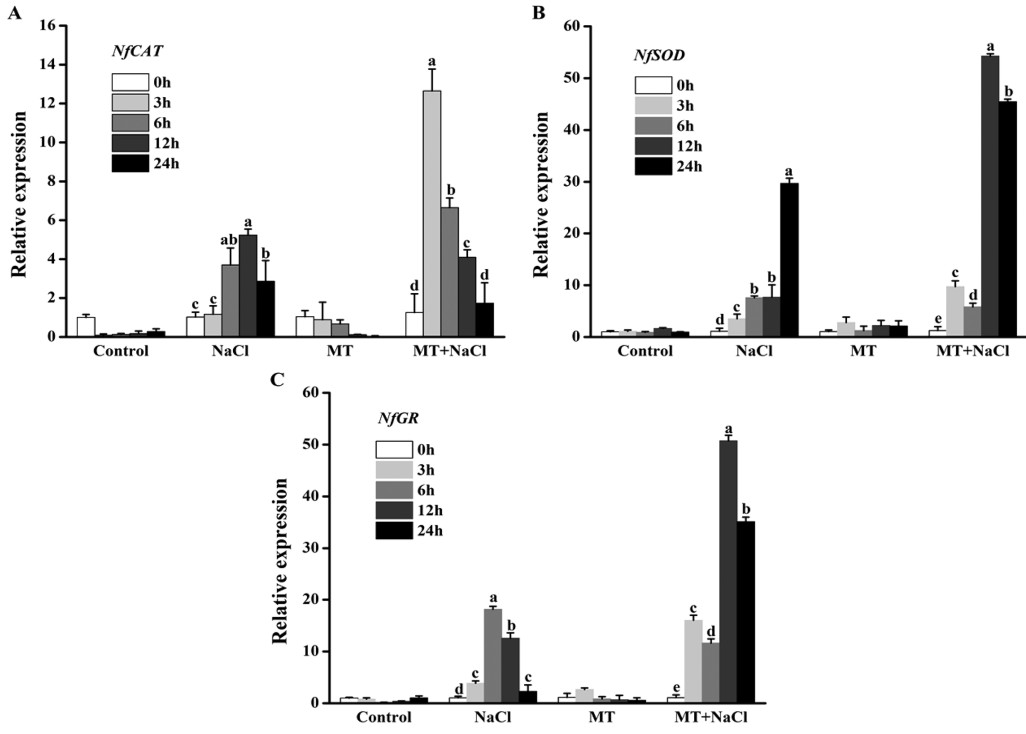

**Figure 7 To further explore the effects of melatonin on the antioxidant defense system of *N. flagelliforme* under salt stress, we investigated the relative expression levels of antioxidant enzymes genes through qRT-PCR.** Effects of melatonin on relative expression levels of (A) *NfCAT*, (B) *NfSOD*, and (C) *NfGR* under 300 mM NaCl stress. Data represent the mean ± SD (*n* = 3). Columns labelled with different letters among treatments show statistical differences (*p* < 0.05).

## Exogenous melatonin counteracts NaCl-induced inhibition of algal cells by regulating biosynthesis of photosynthetic pigments and modulating K⁺ homeostasis

*N. flagelliforme* is a prokaryotic microorganism, which contains a variety of photosynthetic pigments including chlorophyll a (Chl a), phycocyanin (PC), and phycoerythrin (PE). Photosynthesis is the most important physical and chemical process of energy production in *N. flagelliforme*. However, its habitats are often threatened by salt stress (*Jia et al., 2010*). Salt stress can destruct photosynthetic electron transport duo to the production of ROS such as $H_2O_2$ and $^{\bullet}OH$ (*Bose & Howlader, 2020*). It has been proved that MT could protect the photosystem by increasing the activities of antioxidant enzymes to achieve favorable ROS levels (*Yin et al., 2019*). In this study, the obtained results showed that melatonin decreased the content of Chl a, PC, and PE under normal conditions or salinity conditions (Figs. 2A–2C), inconsistent with previous reports in plants under abiotic stresses on the aspect of chlorophyll content (*Arnao & Hernández-Ruiz, 2008*; *Tan et al., 2019*). However, we found that melatonin had a significant improvement on Fv/Fm under salinity conditions (Fig. 2D), which may attribute to the role of melatonin in directly scavenging ROS by boosting antioxidant enzymes activities (Fig. 6) and stimulating related antioxidant genes up-regulation (Fig. 7).

Potassium ($K^+$) is an essential mineral nutrient for algal growth. Inversely, elevated $Na^+$ disturbs ion homeostasis. Organisms respond to salt stress by maintaining low levels of cytosolic $Na^+$ and a high cytosolic $K^+$ (*Golldack et al., 2003*; *Khan et al., 2012*). In this study, salt stress led to a sharp increase in $Na^+$ content and a decrease in $K^+$ content (Figs. 3A and 3B). Exogenous MT alleviated the adverse effects of salt stress by maintaining a higher $K^+$ level or $K^+/Na^+$ ratio (Figs. 3B and 3C). Furthermore, our findings showed that the expression levels of two osmotic genes (*NfEST1* and *NfTrKA*) were individually increased under salt stress after 6 and 12 h by the qRT-PCR assay. However, MT treatment had no significantly difference under non-salinity or salinity conditions (Fig. 4). We deduce the pathway for $K^+$ leak of algal cells under salt conditions *via* ROS-activated $K^+$ channels (*Shabala et al., 2016*). Likewise, we presume the higher $K^+$ content of MT-treated cells under salinity depends on the favorable ROS level. Consistent with this notion was a recent study on rice showed that decrease of salt-induced $K^+$ efflux by exogenous MT was attributed to the ROS scavenge activity (*Liu et al., 2020*).

These above results showed that a higher $K^+/Na^+$ ratio and the good recovery capability of Fv/Fm are crucial for cell survival under salt stress. All these factors together instigated the MT-treated cells under salt stress for better growth due to the improved DW and decreased REL values (Fig. 1).

## Exogenous melatonin improves salt tolerance of algal cells by regulating antioxidant systems

Excessive ROS under salt stress usually causes cellular damage or death. Some reports have revealed that exogenous melatonin can decrease ROS accumulation under salinity stress (*Tan et al., 2000*; *Gao et al., 2019*). In this study, we found algal cells maintained redox homeostasis under normal conditions, MT treatment effectively decreased NaCl-induced $H_2O_2$ and $O_2^{\bullet-}$ content (Fig. 5). As is known from recent evidence, plants or microorganisms efficiently resist oxidative stress *via* a series of ROS-scavenging systems orchestrated by non-enzymatic and enzymatic antioxidant mechanisms (*Arbona et al., 2017*; *Zhao et al., 2020*). During detoxification of oxidative stress, the antioxidant enzyme SOD, is used as the first line of defense to scavenge oxygen free radicals, which dismutates $O_2^{\bullet-}$ to $O_2$ and $H_2O_2$, whereas CAT eliminates $H_2O_2$ by decomposing $H_2O_2$ to $H_2O$. In the ascorbate-glutathione cycle, the enzymatic action of APX can reduce the accumulation of $H_2O_2$. GR plays a key role in the removal of reactive oxygen species in the oxidative stress response. GR catalyzes NADPH to reduce GSSG to produce GSH, which helps to maintain the ratio of GSH/GSSG in the organisms (*Wang et al., 2012*). Thus, the activities of antioxidant enzymes and the redox state of primary antioxidants play important roles in protecting algal cells against free radical damage (*Ding et al., 2018*). As a potent antioxidant, MT acts as a first line of defense against abiotic stress (*Arnao & Hernández-Ruiz, 2014*). It was proved that reduced cellular damage by exogenous MT was closely linked to improved ROS detoxification by the involvement of CAT, SOD, and GR pathway, which play a key role in the upregulation of antioxidant enzymes (*Zhang et al., 2015*). Our results also confirmed this view by measuring antioxidant enzymes activities (Fig. 6). In the present experiment, qRT-PCR trial demonstrated that exogenous melatonin

obviously increased the expression levels of related antioxidant enzymes genes in algal cells under salt stress (Fig. 7).

## CONCLUSIONS

Facing with environmental changes, it is necessary to explore innovative techniques to improve the tolerance of plants or prokaryotic microorganisms against various abiotic stresses. Accumulating evidence indicate that melatonin can enhance salt tolerance associated with scavenging ROS and regulating antioxidant system. In the present study, we for the first time verified the functions of melatonin in eco-friendly terrestrial cyanobacterium against salt stress. Our findings showed that the application of melatonin reduced $H_2O_2$ and $O_2^{\bullet-}$ content, which led to better growth for algal cells, improvement on Fv/Fm, and indirectly maintaining a higher $K^+$ level in algal cells. Meanwhile, the results of antioxidant enzymes activities and antioxidant genes transcription suggested that melatonin enhanced salt tolerance by activating antioxidant system in *N. flagelliforme*. Therefore, the study presented here provides a theoretical basis and technical support for applying melatonin to related fields, especially in crop production and environment protection.

### Funding
This work was supported by the Science and Technology Department of Shaanxi Province, China (2021JQ-544), and by the Shaanxi University of Science & Technology (2019BJ-40). The funders had no role in study design, data collection and analysis, decision to publish, or preparation of the manuscript.

### Grant Disclosures
The following grant information was disclosed by the authors:
Science and Technology Department of Shaanxi Province, China: 2021JQ-544.
Shaanxi University of Science & Technology: 2019BJ-40.

### Competing Interests
The authors declare that they have no competing interests.

### Author Contributions
- Xiaolong Yuan conceived and designed the experiments, performed the experiments, analyzed the data, prepared figures and/or tables, authored or reviewed drafts of the article, and approved the final draft.
- Jing An conceived and designed the experiments, performed the experiments, analyzed the data, prepared figures and/or tables, authored or reviewed drafts of the article, and approved the final draft.
- Tao Zheng performed the experiments, prepared figures and/or tables, and approved the final draft.

- Wenjian Liu performed the experiments, prepared figures and/or tables, and approved the final draft.

## Data Availability

The raw data is available in the Supplemental File.

## Supplemental Information

Supplemental information for this article can be found online at http://dx.doi.org/10.7717/peerj.14479#supplemental-information.

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
