# Peer review of "Exogenous melatonin improves salt tolerance mainly by regulating the antioxidant system in cyanobacterium Nostoc flagelliforme"

_PeerJ, doi:10.7717/peerj.14479_

## Round 0.1 · original submission · Major Revisions

The manuscript was assessed by three experts in this field. One of them is positive about the manuscript content, but the other two raised several concerns about both the manuscript content and presentation.
I consider it appropriate to compare the results with the eukaryotic system, as suggested by Reviewer 2.

Reviewer 1 ·

Basic reporting

no comments

Experimental design

no comments

Validity of the findings

no comments

Additional comments

In the manuscript "Exogenous melatonin improves salt tolerance by mainly regulating the antioxidant system in cyanobacterium Nostoc flagelliforme" The authors observed the effect of melatonin on Nostoc flagelliforme bacteria grown in the presence of NaCl. Different parameters are examined: growth, expression of different antioxidant systems. The results show that there are changes in some of the variables examined. However, there are important points to be clarified in this study.

The melatonin is present in plants and animals, but in which microorganisms is melatonin found? it would be convenient to include bibliography in the introduction.

Other concerns are related to the experimental methodology: why was this concentration of melatonin and NaCl used? were other concentrations tested? is there a dose-response in the parameters tested?

Another concern is the quality (resolution) of the figures which should be improved, since in this manuscript version the figures are small and pixelated.

the presentation of the statistical significance in the figures should be presented in a standard way (with the asterisk system).

The results shown are described in little detail and further analysis of the results would be desirable.

in the discussion it would be desirable to suggest some mechanism of the action of melatonin on the physiology of this bacterium.

in other points:


89 substitute the concentrations of the reagents used in molar concentration

138, 139 144 and 152 Please write what the abbreviation means (SOD CAT APX GR)

171 "After treatment with melatonin, the REL values significantly decreased under salt stress" with NaCl has a value of 100% and with MT+NaCl gives an 85-90% value, is this decrease significant?

174 "cells showed different appearance especially application of melatonin" Please explain why the cells look phenotypically different (figure1 ) since these results do not correspond to the reduction of chlorophyll, phycocyanin or phycoerythrin or to the PSII activity shown in Figure 2.

186 "The melatonin alone improved content of Na + and declined the content of K + compared with the control,respectively." How to explain that on the one hand Na increases and K decreases?


195 please explain in more detail what are the NfEST1 and NfTrKA genes. in figure 4 the expression of these two genes in the presence of MT and MT+Nacl seems to be the same as in the control (the graph is toThe activity of the enzymes increases in the presence of NaCl. MT seems to have an effect on CAT activity (Figure 6A), but on GR activity (Figure 6D) suggests that MT has no effect, but SD in both cases does not seem to correspond to what is indicated (the raw data were impossible for me to understand). However the concentrations of H2O2 and O2 are higher in the presence of NaCl than in the presence of MT+NaCl (figure 5) but the activity of the antioxidant enzymes in the presence of MT+NaCl is higher than with NaCl. how do you explain this result. wouldn't you expect that the higher the activity the lower the amount of oxidizing molecules?

209 The activity of the enzymes increases in the presence of NaCl. MT seems to have an effect on CAT activity (Figure 6A), but on GR activity (Figure 6D) suggests that MT has no effect, but SD in both cases does not seem to correspond to what is indicated (the raw data were impossible for me to understand). However the concentrations of H2O2 and O2 are higher in the presence of NaCl than in the presence of MT+NaCl (figure 5) but the activity of the antioxidant enzymes in the presence of MT+NaCl is higher than with NaCl. how do you explain this result. wouldn't you expect that the higher the activity the lower the amount of oxidizing molecules?


290 "In the present experiment, qRT-PCR trial demonstrated that exogenous melatonin obviously increased the expression levels of related antioxidant enzymes genes inalgalcells under salt stress (Figure 7). It is difficult to compare these qRT-PCR results with those shown in Figures 5 and 6 since these assays were carried out at 14 days, while the qRT-PCR were carried out between 0 and 24 hr (1 day). and obtain some conclusions.

unfortunately I believe that this work is still in a first stage and not yet ready for publication in this journal.

Reviewer 2 ·

Basic reporting

The article entitled “Exogenous melatonin improves salt tolerance by mainly regulating the antioxidant system in cyanobacterium Nostoc flagelliforme” describes the effect of melatonin on the cyanobacteria, Nostoc flagelliforme when exposed to salt stress on the biochemical and physiological level.

The experimental design and data obtained seem to be solid and representative. Unfortunately, the article needs a major revision as this phenomenon has already been reported in eukaryotic algae. I strongly recommend to show the differences and similarities in the physiological, also biochemical, and genetic regulation between prokaryotic and eukaryotic algae, and how the data will add to the academic researchers and biotechnologists regarding novelty.

Experimental design

The manuscript fits the aims and scope of the journal, but lacks novelty.
The methods are described with sufficient detail and information.

Validity of the findings

The data obtained seem to be solid and representative.
The original research question has already been addressed in other publications for eukaryotic algae.

·

Basic reporting

1-Why author used cyanobacterium Nostoc ûagelliforme, what the advantage for using alga
2- Where was isolated Nostoc. flagelliforme

Experimental design

Clear

Validity of the findings

the results are clear

---

## Round 0.2 · accepted · Accept

The reviewers and I assessed your revised version of the manuscript and we think it is now suitable for publication in the Peer Journal.

Reviewer 1 ·

Basic reporting

no comment

Experimental design

no comment

Validity of the findings

no comment

Additional comments

no comment

·

Basic reporting

I accept to publish manuscript in Peer journal

Experimental design

now the manuscript accept

Validity of the findings

Accept